# QUANTIFYING EXPERT SPECIALIZATION FOR EFFECTIVE PRUNING IN MIXTURE-OF-EXPERTS MODELS

## ABSTRACT

Mixture-of-Experts (MoE) architectures enable efficient scaling of language models through sparse activation. However, their deployment is hindered by a significant memory bottleneck, as all expert parameters must remain resident in memory. Expert pruning is an effective technique to mitigate this issue. Existing methods rely on layer-wise metrics based on either routing behavior or expert outputs. These approaches fail to capture the global influence of an expert on cross-layer information flow. In this paper, we introduce a framework for cross-layer information flow analysis. We propose a novel metric called the Expert Specialization Index (ESI). ESI quantifies the entropy of an expert's influence on downstream routing distributions. This allows it to distinguish between functionally specialized experts and redundant, general-purpose ones. Our analysis on Mixtral-8x7B and Qwen1.5-MoE reveals significant differences in their expert specialization profiles. This leads to a key finding we term architecture-strategy fit. Models with highly specialized experts benefit from preserving the original routing distribution via redirection. In contrast, models with less specialized experts are better served by removing experts and re-normalizing routing probabilities.Supported by experimental results, our ESI analysis allows us to explore how to design compression strategies for different MoE architectures. Our findings provide insights into the relationship between model architecture and effective compression strategies.

## 1 INTRODUCTION

Large Language Models (LLMs) have demonstrated remarkable capabilities by scaling to billions or trillions of parameters et al. (2020); OpenAI et al. (2024). However, this success comes with substantial computational and memory costs that pose significant deployment challenges. Mixture-of-Experts (MoE) architectures decouple total parameter count from per-token computation through sparse activation of expert networks Shazeer et al. (2017); Fedus et al. (2022). By selectively activating a subset of experts for each token, MoE models such as Mixtral-8x7B achieve performance comparable to larger dense models at substantially reduced computational cost Jiang et al. (2024). Despite their computational efficiency, MoE models present a significant deployment challenge: all expert parameters must remain resident in GPU memory to accommodate dynamic routing. This memory constraint significantly limits their practical deployment and underscores the urgent need for effective compression techniques Lu et al. (2024).

Existing MoE compression methods often rely on local, layer-wise metrics. Some approaches prune experts based on activation frequency Sarkar et al. (2024), while others merge them based on output similarity Chen et al. (2024). These methods assess an expert's importance from a local perspective. They fail to capture the global influence an expert has on the model's cross-layer information flow. As a result, they do not provide a principled answer to a foundational question: what quantitatively defines an expert's functional importance within the entire network?

Our key insight is that an expert's true importance is determined by how it shapes information flow through subsequent layers, not just by its local properties. To measure this global influence, we introduce a framework for cross-layer information flow analysis. From this, we derive the Expert Specialization Index (ESI), a novel metric that quantifies an expert's "functional focus." ESI measures the entropy of an expert's influence on downstream routing distributions. Low entropy in-

dicates a concentrated influence and high specialization. High entropy suggests a diffuse influence and functional generality. This provides a principled, global perspective.

Application of this framework reveals fundamental differences in expert specialization across MoE architectures. Analysis of Mixtral-8x7B demonstrates low functional differentiation among experts, indicating substantial redundancy. Conversely, architectures such as Qwen1.5-MoE exhibit high specialization with distinct functional roles distributed across experts. These architectural variations motivate the principle of architecture-strategy fit, which posits that optimal compression strategies must align with a model's intrinsic specialization profile. Specifically, highly specialized architectures benefit from compression strategies that preserve routing distributions through expert redirection, whereas architectures with lower specialization achieve better

Our contributions are threefold:

- We introduce the Expert Specialization Index (ESI), a novel metric based on cross-layer information flow entropy. It systematically quantifies an expert's global influence on downstream routing, overcoming the limitations of local signals.
- We characterize expert specialization across different MoE architectures. Our analysis reveals structural differences ranging from "broadly substitutable" to "dominated by critical specialists". This provides new insights into their functional organization.
- We show that aligning the router adaptation strategy with a model's specialization profile achieves state-of-the-art results for training-free expert pruning.

## 2 RELATED WORK

**Mixture-of-Experts Models**  The Mixture-of-Experts (MoE) architecture scales LLM capacity by decoupling total parameters from per-token compute Shazeer et al. (2017); Fedus et al. (2022). In a standard MoE layer, a gating network (router) selects a small subset of expert FFNs (e.g., top-$k$) for each token. Such sparse activation enables models like Mixtral $8\times7$B and Switch Transformers to reach billion- or trillion-parameter scales while activating only a fraction of weights per token, often matching or surpassing dense baselines at comparable FLOPs Jiang et al. (2024); Fedus et al. (2022). A key deployment challenge, however, is memory: to support dynamic routing, all expert parameters are typically resident in device memory during inference, which constrains practicality on commodity hardware Lu et al. (2024). While system-level frameworks improve throughput and training efficiency, they generally do not eliminate the peak memory footprint associated with maintaining all experts Rajbhandari et al. (2022).

**Model Pruning for LLMs**  Model pruning reduces model size by removing redundant weights and is commonly divided into unstructured and structured pruning. Unstructured pruning zeros individual weights and can achieve high sparsity but often requires specialized kernels for efficient inference Hoefler et al. (2021); recent finetuning-free methods for dense LLMs such as SparseGPT and Wanda fall in this category Frantar & Alistarh (2023); Sun et al. (2024). Structured pruning removes entire components (e.g., channels, heads, or blocks) and is typically more hardware-friendly. Our setting aligns with structured pruning at the level of whole experts, which directly reduces the parameter footprint of MoE layers. However, most existing finetuning-free pruning methods target dense architectures and do not account for MoE-specific issues such as routing dynamics and inter-expert dependencies.

**Expert Pruning and Merging**  The compression of MoE models relies on quantifying expert importance, with existing metrics broadly categorized by their information source: router dynamics or expert outputs. Router-centric metrics range from straightforward activation frequency Sarkar et al. (2024); Chen et al. (2022) to more advanced analyses of router parameters, such as weight similarity or finetuning-induced changes Chowdhury et al. (2024); Lee et al. (2024); Xie et al. (2024). While computationally efficient, these methods primarily capture usage patterns, which may not fully reflect an expert's functional contribution. In contrast, output-based metrics offer a more direct measure of function by evaluating expert representations. These methods fall into two categories: reconstruction-based approaches that identify optimal expert subsets, often at a high computational cost (e.g., NAEE Lu et al. (2024)); and similarity-based approaches that merge functionally redundant experts (e.g., HC-SMoE Chen et al. (2024)). However, all these approaches are fundamentally

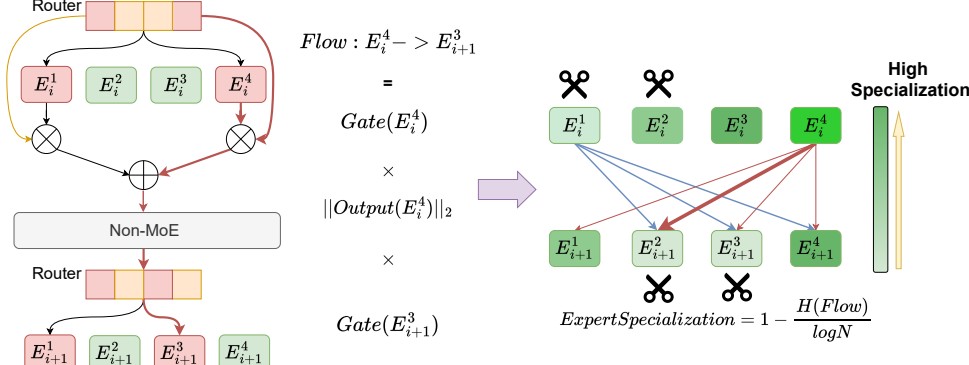

Figure 1: Computation of effective information flowing from $E_i^4$ to $E_{i+1}^3$ during a single forward inference. $N$ is the number of experts

layer-local. Our work addresses this gap by introducing a cross-layer analysis framework that models global dependencies, leading to more robust and effective pruning decisions.

## 3 QUANTIFYING EXPERT SPECIALIZATION

### 3.1 PRELIMINARIES AND PROBLEM FORMULATION

Consider an MoE model with $L$ layers, where each MoE layer $l$ contains $n_e$ expert networks $\{E_i^{(l)}\}_{i=1}^{n_e}$ and a routing function $R^{(l)}$. For an input token representation $\mathbf{x}$, the router computes scores for all experts and selects the top-$k$ experts. The MoE layer output is computed as:

$$\text{MoE}^{(l)}(\mathbf{x}) = \sum_{i \in \text{TopK}(R^{(l)}(\mathbf{x}))} g(E_i^{(l)} \mid \mathbf{x}) \cdot E_i^{(l)}(\mathbf{x}), \tag{1}$$

where $g(E_i^{(l)} \mid \mathbf{x})$ is the normalized gating weight for selected experts.

The expert pruning problem seeks to identify a subset of experts $\mathcal{S}^{(l)} \subset \{1, ..., n_e\}$ to retain in each layer $l$, such that the pruned model maintains performance while reducing parameters. Our goal is to develop a principled importance metric that captures each expert's functional role in the network.

### 3.2 CROSS-LAYER EXPERT FLOW (CLEF)

Traditional importance metrics, focusing on local properties like activation frequency, fail to capture an expert's true impact. In Mixture-of-Experts (MoE) architectures, an expert's influence extends beyond its immediate output—it critically shapes the information landscape for subsequent layers, thereby guiding future routing decisions. To capture this cross-layer dependency, we introduce the Cross-Layer Expert Flow (CLEF) metric.

Our core idea is that a meaningful information flow from a source expert $E_i^{(l)}$ to a downstream expert $E_j^{(l+1)}$ is established only when three sequential conditions are met: (1) the source expert is selected, represented by its gating weight $g(E_i^{(l)} \mid \mathbf{x})$; (2) it produces a significant output, measured by the L2 norm of its output $\|O_i^{(l)}(\mathbf{x})\|_2$; and (3) its influence is received by a downstream expert, captured by the subsequent gating weight $g(E_j^{(l+1)} \mid \mathbf{h}^{(l)}(\mathbf{x}))$. Since these conditions act as sequential gates where the failure of any one nullifies the transmission, we model the flow's magnitude as the product of these three factors. To obtain a stable, global measure, we compute the expected flow over a calibration dataset $\mathcal{D}_{\text{cal}}$:

$$w_{i \to j}^{(l)} = \mathbb{E}_{\mathbf{x} \sim \mathcal{D}_{\text{cal}}} \left[ \underbrace{g(E_i^{(l)} \mid \mathbf{x})}_{\text{Activation}} \cdot \underbrace{\|O_i^{(l)}(\mathbf{x})\|_2}_{\text{Magnitude}} \cdot \underbrace{g(E_j^{(l+1)} \mid \mathbf{h}^{(l)}(\mathbf{x}))}_{\text{Reception}} \right], \tag{2}$$

where $O_i^{(l)}(\mathbf{x})$ is the output of expert $E_i^{(l)}$ and $\mathbf{h}^{(l)}(\mathbf{x})$ is the hidden state after the MoE operation in layer $l$. The overall methodology is conceptually illustrated in Figure 1.

For each expert, we obtain a flow vector

$$\mathbf{Flow}_i^{(l)} = [w_{i \rightarrow 1}^{(l)}, \ldots, w_{i \rightarrow n_e}^{(l)}] \in \mathbb{R}^{n_e}$$

that characterizes its influence distribution over downstream experts. For the final MoE layer $L$, we adapt this concept by computing the flow towards the vocabulary space.

### 3.3 EXPERT SPECIALIZATION INDEX (ESI)

The raw flow vectors, $\mathbf{Flow}_i^{(l)}$, vary in scale across experts, which makes direct comparison difficult. To facilitate a meaningful analysis, we normalize each flow vector into a probability distribution using the softmax function:

$$\mathcal{F}_i^{(l)} = \text{Softmax}(\mathbf{Flow}_i^{(l)}/\tau), \tag{3}$$

where $\tau$ is a temperature parameter that controls the distribution's sharpness. We set $\tau = 1$ in all our experiments. This default setting avoids introducing task-specific hyperparameters and treats the raw flow magnitudes as direct inputs to the probability model.

The entropy of this flow distribution provides insight into an expert's functional role:

$$H_i^{(l)} = -\sum_{j=1}^{n_e} \mathcal{F}_i^{(l)}[j] \log \mathcal{F}_i^{(l)}[j]. \tag{4}$$

A low entropy value indicates a concentrated influence on a few downstream experts, suggesting a specialized function. Conversely, a high entropy value implies a broad influence, indicating a more general-purpose role. To create an interpretable and normalized metric, we define the **Expert Specialization Index (ESI)** on a scale of $[0, 1]$:

$$\text{ESI}_i^{(l)} = 1 - \frac{H_i^{(l)}}{\log n_e}, \tag{5}$$

where $n_e$ is the number of experts in the downstream layer. For the final layer $L$, we use $\log V$ as the normalization factor, where $V$ is the vocabulary size. A higher ESI corresponds to lower entropy and thus greater specialization.

Our central hypothesis is that the ESI serves as a reliable indicator of an expert's functional importance. We posit that experts with a high ESI perform distinct and indispensable functions. In contrast, experts with a low ESI exhibit functional overlap, making them suitable candidates for pruning. We validate this hypothesis in Section 5.

## 4 SPECIALIZATION-GUIDED PRUNING

### 4.1 ESI-GUIDED EXPERT SELECTION

Our pruning strategy is straightforward: for a target pruning ratio $p$, we retain the top $k_l = \lceil (1 - p) \cdot n_e \rceil$ experts with the highest ESI scores in each layer $l$. This is formulated as:

$$\mathcal{S}^{(l)} = \text{TopK}_{\text{ESI}}(\{E_i^{(l)}\}_{i=1}^{n_e}, k_l), \tag{6}$$

where $\text{TopK}_{\text{ESI}}$ is the selection operator. This approach is designed to preserve the most functionally specialized experts—those with high ESI scores—while removing experts with diffuse, redundant information flows (low ESI). The effectiveness of this strategy will be validated in Section 5.

### 4.2 THE CRITICAL CHOICE OF ROUTER ADAPTATION

After selecting experts to prune, adapting the routing mechanism is a critical and under-analyzed decision. We analyze two distinct strategies, visually compared in Figure 2:

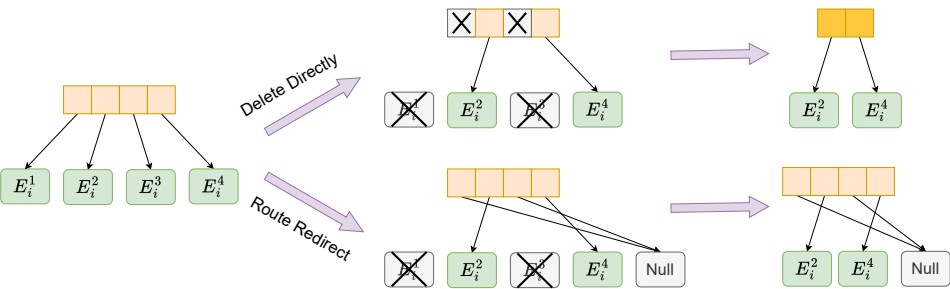

Figure 2: Comparison of router handling strategies after expert pruning. Router Deletion masks pruned expert logits to $-\infty$, forcing renormalization over remaining experts and altering their selection probabilities. Router Redirection preserves the original routing distribution by maintaining all logits while redirecting pruned experts to zero outputs.

**Router Deletion.** This strategy removes pruned experts from routing consideration by masking their logits to $-\infty$ before the softmax operation. The modified router logits.

$$\tilde{R}_i^{(l)}(\mathbf{x}) = \begin{cases} R_i^{(l)}(\mathbf{x}), & \text{if } i \in \mathcal{S}^{(l)} \\ -\infty, & \text{if } i \notin \mathcal{S}^{(l)} \end{cases} \tag{7}$$

This forces a renormalization over the remaining experts, fundamentally altering their selection probabilities but creating a more compact routing space.

**Router Redirection** This strategy preserves the original routing mechanism. The output of the MoE layer is computed by summing only over the set of retained experts, denoted as $\mathcal{S}^{(l)}$:

$$\text{MoE}_{\text{redirect}}^{(l)}(\mathbf{x}) = \sum_{i \in \mathcal{S}^{(l)}} g(E_i^{(l)} \mid \mathbf{x}) \cdot E_i^{(l)}(\mathbf{x})$$

The gating values $g(\cdot)$ are computed using the unmodified router logits over all original experts. If the router selects a pruned expert, its contribution is effectively zero. In the extreme case where all selected experts have been pruned, the input $\mathbf{x}$ passes through as the output. This approach maintains the learned routing dynamics for all experts.

The choice between these two strategies presents a clear trade-off. Router Deletion forces the model to adapt, which is beneficial when experts have overlapping functions. In contrast, Router Redirection prioritizes the preservation of existing routing patterns, which is ideal when experts are highly specialized. As our experiments will demonstrate, the optimal choice is strongly correlated with the model's intrinsic degree of expert specialization.

## 5 EXPERIMENTS

### 5.1 EXPERIMENTAL SETUP

We evaluate our approach on two open-source Mixture-of-Experts (MoE) models with distinct architectures: Mixtral 8x7B Jiang et al. (2024) and Qwen1.5-MoE-A2.7B-Chat Team (2024). Mixtral 8x7B represents a standard MoE design, featuring 8 experts per layer with top-2 routing. In contrast, Qwen1.5-MoE employs a more complex shared-expert architecture with 64 experts and a top-4 activation strategy. These contrasting designs, illustrated in Figure 3, allow us to assess the generalizability and scalability of our method. Additional model details are available in Appendix A.

We compute expert specialization scores using a compact calibration dataset of 64 samples randomly drawn from the C4 corpus Raffel et al. (2020), each standardized to 2048 tokens. This task-agnostic calibration ensures our importance metrics generalize across diverse downstream applications without biasing toward specific domains.

We evaluate the performance of all compressed models on a suite of seven commonsense reasoning benchmarks. We use the lm-evaluation-harness Gao et al. (2021) for this evaluation. The primary

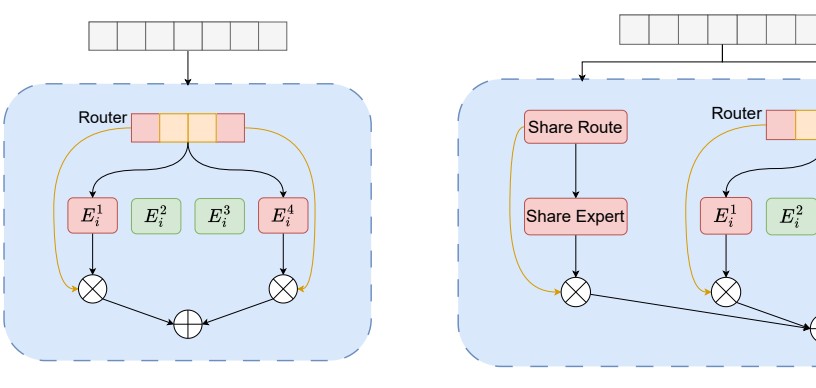

(a) Standard MoE architecture        (b) MoE architecture with shared experts

Figure 3: Comparison of MoE architectures: (a) Standard MoE where all experts are selectively activated based on routing decisions, and (b) MoE with shared experts that are always activated selected experts.

metric is zero-shot accuracy. Our evaluation suite includes ARC-c and ARC-e Clark et al. (2018), BoolQ Clark et al. (2019), HellaSwag Zellers et al. (2019), OpenBookQA Mihaylov et al. (2018), RTE Dagan et al. (2005); Bar-Haim et al. (2006); Giampiccolo et al. (2007); Bentivogli et al. (2009), and Winogrande Sakaguchi et al. (2021).

We compare our specialization-guided approach against several training-free baselines. First, we consider two simple heuristics. **Frequency Pruning** removes experts with the lowest activation counts. **Logit Pruning** removes experts with the lowest average router logit magnitudes. Second, we benchmark against two state-of-the-art methods. **NAEE** Lu et al. (2024) is a pruning method that selects an expert subset to minimize output reconstruction error within each layer. **HC-SMoE** Chen et al. (2024) is a merging method. It first applies hierarchical clustering to group functionally similar experts based on their output similarity, then merges the experts within each resulting cluster.

## 5.2 ANALYSIS OF ARCHITECTURAL DIFFERENCES IN EXPERT SPECIALIZATION

We apply our framework to investigate the intrinsic functional organization of different MoE architectures. Our analysis reveals that Mixtral-8x7B and Qwen1.5-MoE exhibit distinctly different expert specialization patterns.

We employ the Expert Specialization Index (ESI) to characterize specialization for each expert. To highlight the distinction between generalist and specialist experts within each layer, we apply min-max normalization to ESI scores, scaling them to [0, 1]. This assigns a score of 1 to the most specialized expert and 0 to the least specialized expert in each layer. Figure 4 shows the distribution of normalized scores. We provide detailed layer-by-layer heatmaps in Appendix B.

The distributions reveal clear organizational differences. In Mixtral-8x7B (Figure 4a), the boundary between generalist and specialist experts is ambiguous. Most layers show continuous, concentrated ESI distributions, indicating experts share similar specialization levels without clear functional outliers.

Qwen1.5-MoE demonstrates pronounced differentiation between expert types. Figure 4b shows many layers (layers 0, 1, and 2) with ESI distributions heavily skewed towards zero with long upward tails. This pattern indicates large populations of general-purpose experts (low ESI) and small cohorts of highly specialized experts (high ESI). This clear division suggests a more structured functional organization than Mixtral. These architectural insights inform our understanding of compression behavior, which we validate in subsequent sections.

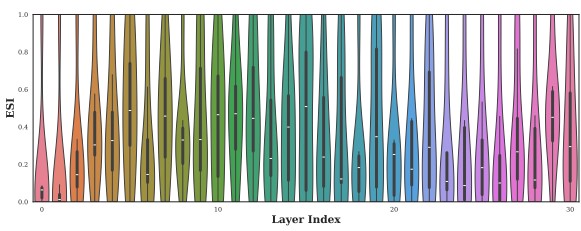 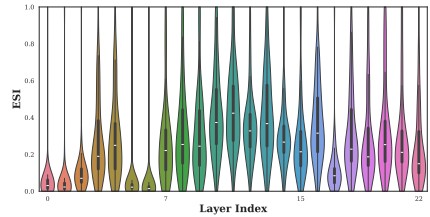

(a) Mixtral-8x7B         (b) Qwen1.5-MoE

Figure 4: Distribution of normalized Expert Specialization Index (ESI) across layers for Mixtral-8x7B and Qwen1.5-MoE. ESI scores are min-max normalized within each layer to a range to highlight the relative specialization of experts. (a) In Mixtral-8x7B, the ESI scores form a continuous and concentrated distribution in most layers, indicating an ambiguous boundary between generalist and specialist experts. (b) In contrast, Qwen1.5-MoE exhibits a pronounced hierarchical structure, where the distribution is heavily skewed towards zero with a long tail.

Table 1: Comparison of router handling strategies at 50% expert pruning. All scores are zero-shot accuracies (%). Best results per model are in **bold**.

| Model | Strategy | ARC-c | ARC-e | BoolQ | HellaSwag | OBQA | RTE | WinoGrande | Avg. |
|-------|----------|-------|-------|-------|-----------|------|-----|------------|------|
| Qwen1.5-MoE | Delete | 30.46 | 62.33 | 66.97 | 49.25 | 27.80 | 56.32 | 63.54 | 50.90 |
| | Redirect | 36.35 | 67.97 | 74.89 | 49.90 | 27.40 | 64.98 | 66.14 | **55.38** |
| Mixtral-8x7B | Delete | 44.88 | 75.55 | 83.30 | 57.22 | 28.20 | 63.95 | 69.69 | **60.38** |
| | Redirect | 25.26 | 50.88 | 43.82 | 34.77 | 19.80 | 52.71 | 53.43 | 40.10 |

## 5.3 VALIDATION OF SPECIALIZATION-GUIDED PRUNING

The architectural differences identified in our analysis lead to a set of clear, testable hypotheses regarding the behavior of these models under compression:

**Hypothesis 1 (Architecture-Strategy Fit)** *The optimal router adaptation strategy is contingent on the model's degree of functional diversity. Models with highly specialized experts will benefit from preserving the original routing distribution (Redirect strategy), while models with functionally redundant experts will benefit from adaptive probability redistribution (Delete strategy).*

**Hypothesis 2 (Importance of Specialization)** *Pruning methods that can accurately identify and preserve functionally specialized experts will show a more significant advantage in models with high functional diversity than in models with high functional overlap.*

We now proceed to validate these hypotheses through task-agnostic pruning on a suite of common-sense reasoning benchmarks.

### 5.3.1 VALIDATION OF THE ARCHITECTURE-STRATEGY FIT PRINCIPLE

To validate **Hypothesis 1**, we compare the performance of the *Delete* and *Redirect* router handling strategies. As presented in Table 1, the results demonstrate a strong dependency on the model's architectural profile. For Qwen1.5-MoE, which is characterized by a distinct cohort of specialized experts, the *Redirect* strategy significantly outperforms the *Delete* strategy, achieving an accuracy 4.48% higher. This architecture necessitates the preservation of the original routing distribution to ensure that its specialists are correctly activated. Modifying the routing probabilities risks disrupting their specialized functions. Conversely, for Mixtral-8x7B, which contains functionally redundant experts, the *Delete* strategy substantially surpasses the *Redirect* strategy, yielding a performance gain of 20.28% . In this architecture, adaptation is more effective than preservation. The *Delete* strategy enables the remaining generalist experts to absorb the workload of pruned peers through probability renormalization. These findings provide strong evidence for our Architecture-Strategy Fit principle: the compression strategy must be aligned with the model's intrinsic functional organization.

Table 2: Task-agnostic pruning results on commonsense reasoning benchmarks at 25% and 50% expert reduction ratios. All scores are zero-shot accuracies (%). Best results per group are in **bold**.

| Model | Method | Ratio | ARC-c | ARC-e | BoolQ | HellaSwag | OBQA | RTE | WinoGrande | Avg. |
|---|---|---|---|---|---|---|---|---|---|---|
| | None | 0% | 41.46 | 73.15 | 79.72 | 57.97 | 30.80 | 67.51 | 69.30 | 59.99 |
| | Frequency | | 38.14 | 67.10 | 69.76 | 54.06 | 29.20 | 58.48 | 68.19 | 54.99 |
| | Logit | | 37.46 | 69.91 | 71.77 | 49.92 | 27.92 | 60.29 | 61.88 | 54.16 |
| | NAEE | 25% | 32.68 | 61.11 | 75.66 | 53.88 | 26.80 | 64.98 | 63.30 | 54.06 |
| | HC-SMoE | | 38.57 | 69.49 | **78.01** | 53.56 | 28.00 | **71.48** | 68.51 | 58.23 |
| Qwen1.5-MoE | **Ours** | | **43.09** | **75.34** | 77.60 | **55.43** | **31.40** | 69.68 | **69.53** | **60.30** |
| | Frequency | | 30.20 | 49.49 | 54.57 | 47.30 | 23.80 | 52.35 | 65.43 | 46.16 |
| | Logit | | 29.52 | 58.88 | 67.46 | 38.92 | 21.80 | 52.71 | 55.80 | 46.44 |
| | NAEE | 50% | 25.68 | 44.49 | 64.96 | 43.51 | 20.20 | 60.65 | 53.75 | 44.75 |
| | HC-SMoE | | 33.36 | 60.77 | **74.89** | 45.23 | 21.60 | 63.18 | 64.01 | 51.86 |
| | **Ours** | | **36.35** | **67.97** | 74.89 | **49.90** | **27.40** | **64.98** | **66.14** | **55.38** |
| | None | 0% | 57.34 | 84.39 | 85.17 | 64.91 | 35.20 | 71.12 | 76.56 | 67.81 |
| | Frequency | | 43.94 | 72.98 | 70.92 | 57.07 | 30.80 | 57.40 | 74.11 | 58.17 |
| | Logit | | 48.46 | 78.75 | 78.96 | 60.58 | 30.80 | 62.82 | 74.51 | 62.13 |
| | NAEE | 25% | 50.88 | **81.07** | 83.82 | **61.80** | **31.40** | 63.68 | 75.06 | 63.96 |
| | HC-SMoE | | **51.37** | 80.47 | **85.44** | 61.41 | 30.80 | **65.71** | 75.16 | **64.34** |
| Mixtral-8x7B | **Ours** | | 50.43 | 79.84 | 85.29 | 61.04 | 31.20 | 63.54 | **75.22** | 63.79 |
| | Frequency | | 42.92 | 73.06 | 74.80 | 53.34 | 26.80 | 55.96 | 69.46 | 56.62 |
| | Logit | | 41.38 | 73.57 | 68.87 | 57.03 | 27.20 | 59.21 | 72.14 | 57.06 |
| | NAEE | 50% | **45.55** | 74.07 | 81.01 | 56.76 | **28.40** | 64.24 | 72.01 | 60.15 |
| | HC-SmoE | | 45.22 | 74.62 | 82.15 | 57.18 | 27.20 | **64.60** | **72.69** | **60.52** |
| | **Ours** | | 44.88 | **75.55** | **83.30** | 57.22 | 28.20 | 63.95 | 69.69 | 60.38 |

### 5.3.2 VALIDATION OF IMPORTANCE OF SPECIALIZATION

We conduct task-agnostic pruning experiments using the settings described in Section 5.1. Since NAEE's search space is too large on Qwen, we limit its search to 1,000 iterations. We apply the Redirect strategy for Qwen and the Delete strategy for Mixtral. Table 2 presents the main results of our specialization-guided pruning compared to strong baselines.

On Qwen1.5-MoE, a model we identified as having high functional diversity, our method achieves substantial improvements over all baselines. At 25% pruning, our approach achieves an average accuracy 0.31% higher than the original model and outperforms the best baseline, HC-SMoE, by 2.07%. At 50% pruning, this advantage increases to 3.52%. This demonstrates the critical importance of accurately identifying and preserving highly specialized experts in such architectures. Our method shows even greater advantages at higher compression ratios.

Conversely, on Mixtral-8x7B, which our analysis characterized as having high functional overlap, the performance gap between different methods is minimal. Our method outperforms Frequency and Logit pruning but performs slightly below state-of-the-art methods. This result is consistent with our hypothesis: when experts are largely redundant and substitutable, the specific choice of pruning metric has a less pronounced impact on performance. Section 5.4 provides further evidence of expert substitutability in Mixtral. These results provide support for **Hypothesis 2**. Furthermore, we conducted a task-specific pruning analysis, with detailed results presented in Appendix C.

### 5.4 MORE ANALYSIS

**Corroborating Analysis via Functional Similarity** To further validate our findings on architectural specialization, we conducted a complementary analysis based on the functional similarity of experts. We computed the cosine similarity of the average output vectors for experts within each layer, providing a direct measure of their functional overlap.

As shown in Figure 5, the results strongly corroborate our ESI-based conclusions. The experts in Mixtral-8x7B (Figure 5a) exhibit high inter-expert similarity, confirming that they have low functional differentiation and are largely interchangeable. In contrast, the experts in Qwen1.5-MoE (Figure 5b) show significantly lower similarity to one another, which aligns with our finding that this architecture fosters a high degree of functional specialization. This analysis provides an alter-

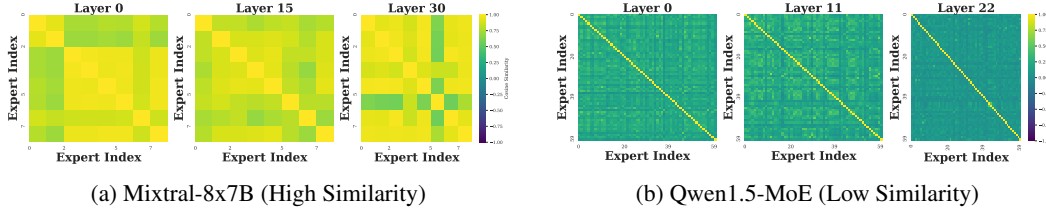

(a) Mixtral-8x7B (High Similarity)    (b) Qwen1.5-MoE (Low Similarity)

Figure 5: Intra-layer expert output cosine similarity analysis. Heatmaps show the similarity matrix between all experts within selected layers.(a) Mixtral-8x7B exhibits high similarity among experts.(b) Qwen1.5-MoE demonstrates low similarity, suggesting greater expert diversity.

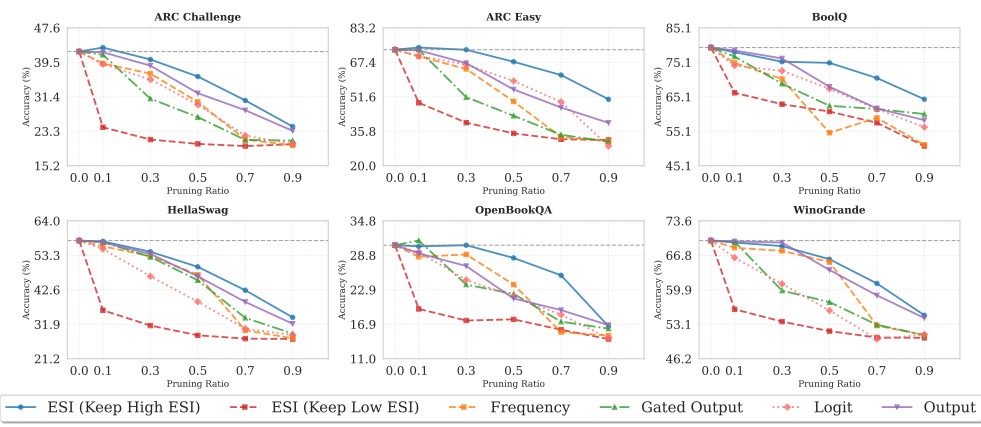

Figure 6: Ablation study on the importance of cross-layer information. Pruning based on our full ESI metric consistently outperforms layer-local baselines.

native yet consistent perspective, reinforcing the conclusion that Mixtral's experts are more easily substitutable than Qwen's.

**Validating the Cross-Layer Approach**    To isolate the contribution of the downstream gating term $(g(E_j^{(l+1)}))$, we compare our full ESI metric against two layer-local baselines: **Output L2** (output magnitude only) and **Gated Output L2** (local gating + magnitude). Figure 6 shows ESI-based pruning consistently outperforms both local variants.

We validate ESI's effectiveness through inverse pruning. Removing the highest ESI experts causes severe accuracy degradation. Removing just 10% of top experts on ARC-C results in an 18.77% accuracy drop, confirming these experts are functionally critical and irreplaceable. This demonstrates that downstream information reception is essential for accurate expert importance assessment.

# 6    CONCLUSION

We introduced the Expert Specialization Index (ESI), a new metric to measure an expert's functional role based on its cross-layer influence. Our method moves beyond traditional, layer-local heuristics. Our ESI-guided pruning strategy achieves state-of-the-art, training-free compression. It also serves as an analytical tool to investigate the internal structure of MoE models. Using ESI, we identified a fundamental difference between MoE architectures. Some models, like Qwen1.5-MoE, show high functional diversity, while others, like Mixtral-8x7B, have high functional overlap. This finding led us to propose the "architecture-strategy fit" principle. This principle states that the optimal compression strategy depends on the model's underlying specialization pattern. This work provides a dual contribution: a practical, high-performance pruning method and a deeper, more structured understanding of the functional organization of MoE models.

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

## A    MODEL DETAIL

Our experiments are conducted on two prominent open-source Mixture-of-Experts (MoE) models, chosen for their distinct architectural designs which allow us to study varying degrees of expert specialization. The key specifications are summarized in Table 3.

Table 3: Key architectural specifications of the evaluated MoE models.

| Parameter | Mixtral-8x7B | Qwen1.5-MoE-A2.7B |
|---|---|---|
| Total Parameters | 47B | 14.3B |
| Active Parameters | 13B | 2.7B |
| Total Experts per Layer | 8 | 64 |
| – Routed Experts | 8 | 60 |
| – Shared Experts | 0 | 4 |
| Top-K Routing | 2 | 4 |

## B    DETAILED ESI SCORE HEATMAPS

To complement the violin plots in Section 5.2, which show the overall distribution of specialization, this appendix provides detailed heatmaps of the normalized ESI scores. These visualizations allow for a fine-grained inspection of the specialization of each individual expert across all MoE layers.

Figure 7 and Figure 8 present these heatmaps for Mixtral-8x7B and Qwen1.5-MoE, respectively. The heatmap for **Mixtral-8x7B** (Figure 7) confirms that its experts have a low degree of functional differentiation. The ESI distribution within each layer is relatively uniform, as indicated by the lack of extreme color contrasts. This visual evidence supports the conclusion that most experts share a similar, moderate level of specialization.

In contrast, the heatmap for **Qwen1.5-MoE** (Figure 8) reveals that some layers exhibit a particularly stark differentiation in expert capabilities. This is especially evident in the first three layers (rows 0-2), where the vast majority of experts are generalists (dark cells), while only one or two experts per layer are highly specialized, achieving a maximum ESI score of 1.0 (intensely bright cells). This stark visual contrast provides further evidence for the clear division of labor and the presence of a few critical specialists within a large pool of generalists in the Qwen1.e-MoE architecture.

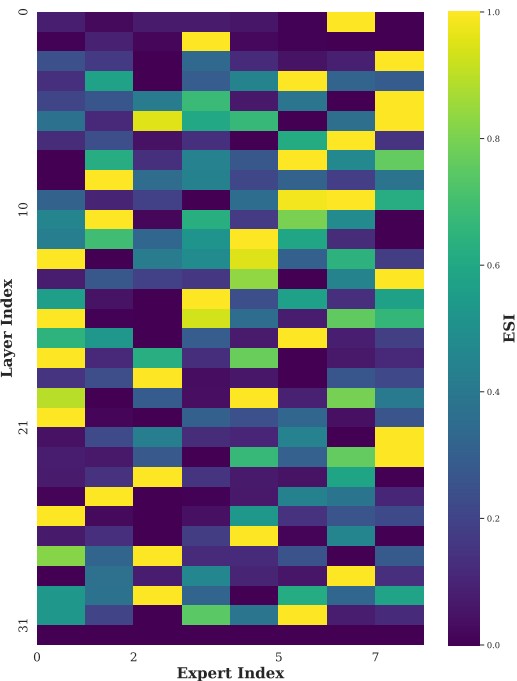

Figure 7: Heatmap of layer-wise normalized Expert Specialization Index (ESI) scores for all experts inMixtral-8x7B. Brighter colors indicate higher relative specialization within that layer.

## C    ANALYSIS OF TASK-SPECIFIC SPECIALIZATION

We conduct a task-specific pruning analysis. In this scenario, we calibrate the model directly on the training or validation set of a target task to identify experts that are functionally specialized for that domain. We focus this analysis on Qwen1.5-MoE, as its larger expert pool (64 experts per layer) offers a finer-grained potential for domain specialization.

We select three diverse domains for this study: commonsense reasoning with HellaSwag Zellers et al. (2019), mathematical problem-solving with MathQA Amini et al. (2019), and medical knowledge with MedMCQA Pal et al. (2022). Table 4 presents the 5-shot accuracy at a 50% pruning ratio, comparing task-specific calibration against a generic C4 calibration.

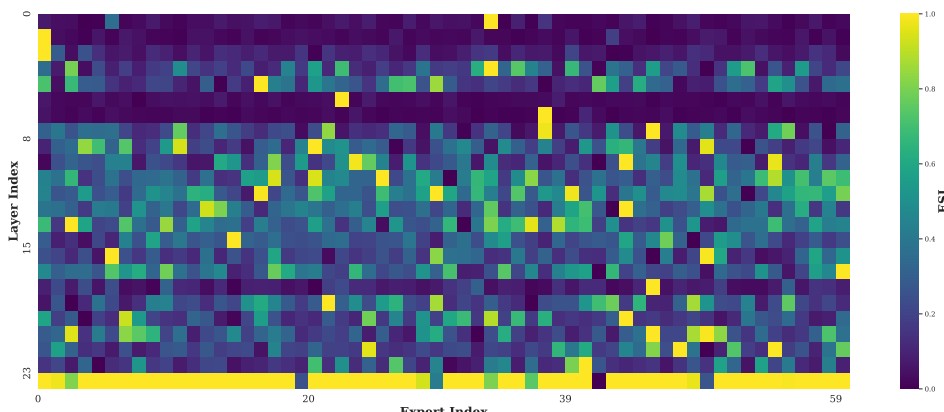

Figure 8: Heatmap of layer-wise normalized Expert Specialization Index (ESI) scores for all experts in Qwen1.5-MoE.

Table 4: Task-specific pruning results on Qwen1.5-MoE at 50% expert reduction. Accuracy (%) with 5-shot evaluation. Best results per task are in **bold**.

| Method | Calibration Data | HellaSwag | MathQA | MedMCQA |
|---|---|---|---|---|
| None | - | 57.97 | 35.75 | 49.58 |
| HC-SMoE | C4 | 45.23 | 26.50 | 34.74 |
| | HellaSwag | 46.00 | 26.06 | 33.21 |
| | MathQA | 40.01 | **34.64** | 35.48 |
| | MedMCQA | 43.58 | 32.70 | 38.99 |
| **Ours** | C4 | 49.65 | 31.39 | 37.25 |
| | HellaSwag | **50.18** | 26.87 | 36.27 |
| | MathQA | 47.98 | 32.79 | 36.24 |
| | MedMCQA | 46.31 | 30.45 | **45.23** |

The results reveal how expert organization varies across different knowledge domains. Our specialization-guided approach demonstrates a significant advantage in domains with highly structured and specialized knowledge. On MedMCQA, calibrating on in-domain data yields a remarkable 45.23% accuracy, outperforming the best baseline (HC-SMoE calibrated on MathQA) by over 6 percentage points. This strongly suggests that specialized medical knowledge is concentrated within a specific subset of experts, which our cross-layer flow analysis can precisely identify and preserve.

Conversely, the outcomes on other domains highlight the nuances of expert contributions. For a general-knowledge task like HellaSwag, in-domain calibration provides only modest gains over C4 calibration, indicating that commonsense reasoning relies on a broader, more distributed set of experts rather than a small, highly specialized cohort. The results on MathQA reveal a potential limitation: the similarity-based merging of HC-SMoE proves more effective. This suggests that complex, multi-step reasoning may depend more on the collaborative function of multiple generalist experts, a scenario where preserving a few specialists is less critical than maintaining the functional diversity of the entire expert pool.

## D    THE USE OF LARGE LANGUAGE MODELS

During the preparation of this manuscript, we utilized Large Language Models (LLMs) as a writing assistant. LLMs helped summarize related literature and refine the manuscript's clarity and grammar. All core ideas, experimental designs, analyses, and conclusions are the original contributions of

the human authors. LLMs served solely as tools to enhance presentation quality without contributing to the scientific discoveries themselves.

