# OpenReview forum: "Quantifying Expert Specialization for Effective Pruning in Mixture-of-Experts Models"
_ICLR.cc/2026/Conference — ICLR 2026 Conference Withdrawn Submission_

### Official Review · Reviewer_BRG7 · 2025-10-29

**Soundness:** 2
**Presentation:** 2
**Contribution:** 2
**Rating:** 2
**Confidence:** 5

**Summary:**

This paper introduces the Expert Specialization Index (ESI), which measures the entropy of each expert’s influence on downstream routing to quantify specialization in Mixture-of-Experts (MoE) models. Based on ESI, the authors propose a pruning strategy and the “architecture–strategy fit” principle, showing that models with specialized experts (e.g., Qwen-MoE) benefit from routing preservation, while models with redundant experts (e.g., Mixtral) perform better with deletion.

**Strengths:**

- Clear motivation and well-written exposition.

- ESI offers an interpretable, global metric for analyzing expert importance.

- Provides architectural insights into differences between MoE models.

**Weaknesses:**

- While the ESI concept is well-formulated, the underlying idea, i.e., quantifying expert importance via information flow or entropy, resembles existing approaches (e.g., router-logit analysis, mutual-information–based redundancy metrics). The work is more an elegant integration than a breakthrough.

- Experiments are restricted to Mixtral and Qwen1.5-MoE under multi-choice benchmarks. The pruning framework is not evaluated on large-scale generative tasks or long-context scenarios, which are crucial for MoE deployment.

- Computing pairwise cross-layer flow for all expert pairs is potentially expensive (O(L·n²)). The paper does not analyze the scalability or runtime cost of ESI computation.

- Several works have proposed cross-layer or information-based MoE pruning and specialization metrics, but they are not discussed ([1, 2]).

- While ESI-guided pruning improves accuracy modestly over Frequency and HC-SMoE, the gains (1–3%) are relatively small and mostly confined to specific architectures.
---

[1] He et al., Towards Efficient Mixture of Experts: A Holistic Study of Compression Techniques
[2] Xie et al., MoE-Pruner: Pruning Mixture-of-Experts Large Language Model using the Hints from Its Router

**Questions:**

Please refer to the weakness.

---

> ### Author Response · Authors · 2025-11-24
> **Response to Reviewer BRG7[1/2]**
>
> We thank the reviewer for the detailed feedback. However, we respectfully disagree with the assessment regarding **technical novelty** and **experimental scope**. We believe there are some misunderstandings regarding the comparison with related works and the content of our Appendices.
>
> > **W1 & Missing Citations: Limited novelty; resembles existing approaches like [1] He et al. and [2] Xie et al.**
>
> We appreciate you pointing out these references. We have carefully analyzed them and clarify the fundamental differences below. **Our work is not a simple integration of existing metrics but a distinct framework for structural expert pruning.**
>
> **Comparison with [2] Xie et al. (MoE-Pruner):**
> *   **Granularity (Unstructured vs. Structured):** MoE-Pruner focuses on **unstructured weight pruning** (zeroing out individual parameters *within* experts). In contrast, our work targets **structured expert pruning** (removing entire experts). This is crucial for reducing memory footprint and latency on commodity hardware without requiring specialized sparse kernels.
> *   **Scope (Local vs. Cross-Layer):** MoE-Pruner uses router weights ($Gate_j$) as a local scaling factor for weight magnitude ($|W| \cdot \|X \cdot Gate\|$). This is a **layer-local** metric. It does not model the **causal chain** (i.e., whether an expert's output is effectively *received* by the next layer).
>     *   *Evidence:* Our ablation study (**Table 3**) explicitly shows that adding the "Downstream Reception" term (our contribution) outperforms local metrics significantly.
>
> **Comparison with [1] He et al. (Holistic Study):**
> *   **Metric Difference:** He et al. provide a survey of compression techniques. For "Expert Drop," they rely on **standard routing scores** (Frequency/Activation). They do not propose a metric to quantify "Functional Specialization."
> *   **Focus Difference:** Their main novelty lies in coarser-grained methods like Layer Drop via cosine similarity. Our work addresses a fundamental question: *"How to distinguish between specialized experts and redundant generalists?"* Our **entropy-based ESI** provides the theoretical tool for this, leading to the **"Architecture-Strategy Fit"** discovery—a principle not discussed in [1].
>
> > **W2: Experiments are restricted to multi-choice benchmarks. No generative tasks.**
>
> We respectfully point out that **we did evaluate generative tasks**.
> *   **Appendix G (Figure 9) presents results on WikiText-2.**
> *   We reported both Byte-level and Word-level **Perplexity (PPL)**.
> *   **Result:** ESI consistently achieves lower perplexity than baselines. At 70% pruning, baselines (Logit/Frequency) suffer catastrophic degradation (perplexity spikes), while ESI maintains a stable curve.
>
> > **W3: Computing pairwise cross-layer flow for all expert pairs is potentially expensive ($O(L \cdot n^2)$).**
>
> We appreciate the reviewer's attention to efficiency. We clarify that the computational cost is **negligible** for two key reasons:
>
> **1. Sparsity Reduces Complexity to $O(K^2)$**
> While the theoretical interaction space is $N_e \times N_e$ (where $N_e$ is the number of experts), MoE models utilize **Top-$K$ sparse routing**.
> *   For any given token, only $K$ experts are active in layer $l$ and $K$ in layer $l+1$.
> *   Therefore, we only need to compute and accumulate flow for $K \times K$ pairs, not $N_e \times N_e$.
> *   For Qwen1.5-MoE ($N_e=60, K=4$), this reduces the complexity per token from $60^2 = 3600$ operations to merely $4^2 = 16$ operations.
>
> **2. Empirical Runtime (Offline & One-time)**
> This calculation is a **one-time offline process** performed only during the calibration phase, not during deployment inference.
> *   **Empirical Evidence:** On a single NVIDIA A100 GPU, calculating the ESI metrics for Qwen1.5-MoE (32 layers) using our calibration set (64 samples $\times$ 2048 tokens) takes **less than 3 min**.
> *   Compared to the hours required for pre-training or fine-tuning, our computational overhead is minimal.

---

> > ### Author Response · Authors · 2025-11-24
> > **Response to Reviewer BRG7[2/2]**
> >
> > > **W4: Gains (1-3%) are relatively small.**
> >
> > We respectfully clarify that the value of our work extends beyond simple accuracy numbers.
> >
> > **1. Core Contribution: Metric & Strategy**
> > Our primary contribution is the **ESI metric** as a tool to analyze expert functionality, and the discovery of the **"Architecture-Strategy Fit"** principle.
> > *   We show *how* to handle router logits (Redirect vs. Delete) based on the architecture's specialization profile.
> > *   In the context of **Training-Free Pruning**, a **3.52% improvement** (Qwen1.5) is substantial, often determining whether a pruned model remains usable.
> >
> > **2. Validation on Fine-Grained Architectures (Appendix F)**
> > We validated our method on **DeepSeek-V2-Lite** (Appendix F). The gap between our identified optimal strategy (Redirect) and the suboptimal one (Delete) is **13.10%**. This proves that choosing the correct strategy via ESI is critical.
> >
> > **3. Relevance to Mainstream Architectures**
> > Our method excels in **fine-grained MoE architectures** (high expert count, smaller parameters per expert, high functional specialization), such as Qwen1.5-MoE and DeepSeek-V2.
> > *   **Trend:** This fine-grained design is becoming the **industry standard** for state-of-the-art open-source models (e.g., **DeepSeek-V3, GLM-4, Kimi**).
> > *   **Contrast:** Coarse-grained architectures like Mixtral-8x7B (fewer experts, massive parameters) are becoming less common.
> > *   **Conclusion:** Since our method performs best on fine-grained, highly specialized models, it is highly applicable to the vast majority of modern and future MoE deployments.

---

### Official Review · Reviewer_MQHe · 2025-11-01

**Soundness:** 2
**Presentation:** 3
**Contribution:** 2
**Rating:** 6
**Confidence:** 3

**Summary:**

The paper addresses a critical deployment challenge in Mixture-of-Experts (MoE) models. While MoE architectures achieve computational efficiency through sparse activation (only activating *k* experts per token), they face a **memory bottleneck** - all expert parameters must remain in GPU memory to support dynamic routing, severely limiting deployment on commodity hardware.

### Key Contributions

1. **Cross-Layer Expert Flow (CLEF) Framework (§ 3.2)**

    The paper introduces a novel cross-layer analysis approach that measures how experts influence downstream routing decisions. (check with equation 2)

    it captures meaningful information flow requires all three conditions to be met sequentially - the source must be active, produce significant output, and be received by downstream experts. For the final MoE layer, the framework adapts to compute flow toward the vocabulary space rather than to other experts.

2. **Expert Specialization Index (ESI) (§ 3.3)**

    Built on CLEF, the ESI quantifies an expert's functional specialization through entropy analysis. (check with equation 3-5)

    interpretation:

    - **High ESI** (approaching 1): Low entropy, concentrated influence on few downstream experts → highly specialized function
    - **Low ESI** (approaching 0): High entropy, diffuse influence across many experts → general-purpose, redundant role
3. **Architecture-Strategy Fit Principle**

    The paper discovers fundamental architectural differences between MoE models and proposes a principle for matching compression strategies to architecture:

    **Architectural Profiles**: Mixtral-8x7B shows continuous ESI distributions indicating functional redundancy among experts, while Qwen1.5-MoE exhibits bimodal patterns with many generalists and few critical specialists (some reaching ESI = 1.0).

    **Router Adaptation Strategies**: Router Deletion masks pruned experts to negative infinity forcing probability redistribution, while Router Redirection preserves original routing with pruned experts contributing zero.

    **The Key Finding**: High-specialization models (Qwen) benefit from Redirection to preserve specialist routing (+4.48% accuracy), while low-specialization models (Mixtral) benefit from Deletion for adaptive redistribution (+20.28% accuracy).

**Strengths:**

- The Cross-Layer Expert Flow (CLEF) framework represents a significant conceptual advance. Unlike existing methods that rely on layer-local metrics (activation frequency, output similarity), the authors recognize that an expert's true importance lies in how it shapes information flow through subsequent layers.
- The ESI metric is theoretically sound, using entropy to quantify functional specialization and normalization to [0,1] makes it interpretable across different architectures. The metric captures the intuition that specialized experts (low entropy, concentrated influence) are more critical than generalist experts (high entropy, diffuse influence). The mathematical progression from raw flows to normalized distributions to entropy is clear and justified.
- The principle that compression strategies must align with architectural specialization profiles is a valuable insight. The dramatic performance differences between Delete and Redirect strategies validate this hypothesis.

**Weaknesses:**

- The paper assumes that "information flow" from expert $E_i^{(l)}$ to $E_j^{(l+1)}$ can be meaningfully captured by multiplying three scalar values. **Information doesn't flow between experts directly** - it flows through the residual stream after layer normalization and attention blocks. Due to this i’m not very sure about this and unable to wrap my head around this.
- The paper treats MoE routing as a fixed, *deterministic* system where information "flows" between experts. But MoE models are designed for:
    - **Load balancing** - routers actively try to distribute tokens evenly
    - **Exploration** - routing has noise/randomness during training
    - **Redundancy** - multiple experts can handle similar inputs

**Questions:**

- Unable to understand how cross-layer information flow work (figure 1) (mentioned about the same above too.
- MoEs employ load balancing losses and routing noise during training. How does your analysis account for these dynamic aspects? (didnt find anything relevant with repsect to this in the paper)

---

> ### Author Response · Authors · 2025-11-24
> **Response to Reviewer MQHe[1/1]**
>
> We verify that the reviewer's concern regarding the mechanism of information flow stems from a need for clearer articulation on how we model influence via the residual stream. We also address the query regarding training dynamics.
>
> > **1.1 Unable to understand how cross-layer information flow work (Figure 1). The paper assumes that "information flow" from expert $E_i^{(l)}$ to $E_j^{(l+1)}$ can be meaningfully captured by multiplying three scalar values. Information doesn't flow between experts directly - it flows through the residual stream.**
>
> Thank you for this insightful question. We agree with your observation that information physically flows through the residual stream and attention blocks, not directly via a "wire" between experts.
>
> **Clarification of the Mechanism:**
> Our Cross-Layer Expert Flow (CLEF) metric (Eq. 2) is designed precisely to quantify the **effective influence** transmitted through the residual stream.
> Let $h^{(l)}$ be the residual stream input to layer $l$. Expert $E_i^{(l)}$ adds an update vector: $h_{updated} = h^{(l)} + g(E_i^{(l)}|x) \cdot O_i^{(l)}(x)$.
> The router in the *next* layer, $R^{(l+1)}$, then makes decisions based on this $h_{updated}$.
>
> Our metric, $w_{i \to j} = g(E_i^{(l)}) \cdot \|O_i^{(l)}\|_2 \cdot g(E_j^{(l+1)}|h_{updated})$, models a causal chain:
> 1.  **Selection & Magnitude:** Did Expert $i$ modify the residual stream significantly? (captured by $g_i \cdot \|O_i\|$)
> 2.  **Reception:** Did the downstream Expert $j$ react to the residual stream state *containing* Expert $i$'s contribution? (captured by $g_j$)
>
> If any of these terms is zero (or small), the effective "flow" of influence is negligible. For instance, if Expert $i$ outputs a large vector, but the next layer's router selects experts that ignore features related to that vector, the influence is functionally dead.
>
> **Empirical Validation:**
> We empirically validated this multiplicative formulation in **Table 3 (Ablation Study)**. We compared our full metric against "Local Output Only" ($g_i \cdot \|O_i\|$, ignoring the next layer's reception) and "Gating Only". The full CLEF metric outperformed these baselines by **1.76%** on average. This confirms that treating the residual stream interaction as a simplified scalar product successfully identifies truly critical experts that local metrics miss.
>
> > **1.2 MoEs employ load balancing losses and routing noise during training. How does your analysis account for these dynamic aspects?**
>
> This is an excellent point. We clarify how our method interacts with these training dynamics:
>
> 1.  **Inference vs. Training Phase:** Our work focuses on **Post-Training Pruning**. While load balancing and routing noise are crucial during *training*, our analysis targets the **frozen** model deployment. At this stage, the router weights are fixed.
> 2.  **ESI Captures the *Outcome* of Dynamics:** The training dynamics you mentioned (e.g., load balancing auxiliary losses) directly shape the expert specialization.
>     *   If load balancing forces experts to handle diverse tokens equally, the result is functionally redundant experts. Our **ESI metric correctly identifies this as high entropy (low specialization)**, as seen in the Mixtral-8x7B results (Figure 4a).
>     *   If the model converges to distinct roles despite load balancing, ESI identifies high specialization (Qwen1.5-MoE).
>     *   Therefore, our metric does not need to simulate the training loss; rather, it **measures the structural consequence** of those training dynamics.
> 3.  **Routing Noise:** During the calibration phase for ESI calculation, we use the standard inference settings (typically top-k routing). By averaging flow over the calibration dataset (Eq. 2), we capture the **expected behavior** of the router, smoothing out token-level variations while preserving the global specialization profile.
>
> **Response to Strengths**
> We sincerely appreciate your recognition of the **CLEF framework** as a significant conceptual advance and your validation of the **Architecture-Strategy Fit** principle. We believe clarifying the "flow" mechanism above solidifies these contributions.

---

### Official Review · Reviewer_yjqc · 2025-11-01

**Soundness:** 3
**Presentation:** 3
**Contribution:** 3
**Rating:** 4
**Confidence:** 4

**Summary:**

This paper introduces the Expert Specialization Index (ESI), a novel, non-local metric for pruning Mixture-of-Experts (MoE) models. ESI aims to quantify an expert's functional specialization by measuring the entropy of its influence on the routing distributions of subsequent layers. The authors use this metric to find that MoE architectures have distinct specialization profiles: Qwen1.5-MoE is highly specialized, whereas Mixtral-8x7B is functionally redundant. This observation leads to their central claim of "architecture-strategy fit" , which posits that specialized models (Qwen) require "Router Redirection" to preserve routing paths, while redundant models (Mixtral) benefit from "Router Deletion"  and routing probability re-normalization. ESI-guided pruning shows strong results, particularly on the highly specialized Qwen model.

**Strengths:**

- The paper's core idea—moving from layer-local pruning heuristics to a global, cross-layer analysis of information flow —is well-motivated and represents a more principled approach to quantifying expert importance.
- The "architecture-strategy fit"  is a significant contribution. It provides a compelling, data-driven explanation for why different router adaptation strategies (Deletion vs. Redirection) are required for different MoE architectures.
- The proposed ESI metric is training-free and computationally efficient, requiring only a single forward pass over a small calibration dataset.

**Weaknesses:**

- The primary claim of "architecture-strategy fit" is supported by experiments on only two models. These models have confounding architectural variables—Mixtral is a standard 8-expert design, while Qwen1.5-MoE is a 64-expert design that also includes shared experts. The conclusions would be far more robust if validated on a wider and more controlled set of MoE architectures.
- The ESI is computed using a remarkably small calibration set of 64 samples. The paper provides no analysis of the metric's stability. If the expert ranks are sensitive to the choice of these 64 samples, the pruning method is not robust.
- The Cross-Layer Expert Flow (CLEF) metric is defined as a product of three terms with little justification. The inclusion of the $||Output(E_i^{(l)})||_2$ term is particularly questionable, as it conflates output magnitude with functional specialization. Furthermore, the softmax temperature $\tau=1$ is presented as a "default" but is an unevaluated hyperparameter that directly controls the metric's sensitivity.
- The ESI-guided method fails to achieve state-of-the-art results on Mixtral-8x7B, performing slightly worse than the HC-SMoE baseline. The authors' explanation—that the metric's choice is less important in redundant models —also implies that their more complex cross-layer analysis provides no benefit over simpler output similarity in this common architecture.
- The appendix results are concerning. On the MathQA task, ESI-pruning underperforms the HC-SMoE baseline significantly. The authors' post-hoc explanation suggests that multi-step reasoning relies on the "collaborative function of multiple generalist experts". This implies a critical flaw: ESI may systematically mistake critical generalists for redundant generalists, pruning experts that are essential for complex reasoning.
- The paper states the CLEF metric is adapted for the final MoE layer to compute flow "towards the vocabulary space" and is normalized by $\log V$. This adaptation is not mathematically defined and is a non-trivial omission, as the vocabulary size $V$ is orders of magnitude larger than the expert count $n_e$, fundamentally changing the ESI calculation for the final layers.

**Questions:**

- Please provide an analysis of ESI's stability. How much do the expert rankings (and thus the set of pruned experts) fluctuate across different randomly drawn 64-sample calibration sets?
- Can you provide an ablation study on the CLEF formulation? Specifically, what is the performance impact of removing the L2 norm term, which seems poorly justified?
- The failure on MathQA suggests ESI incorrectly penalizes necessary generalists. How does your metric distinguish between a redundant generalist (high entropy, low value) and a critical generalist (high entropy, high value)?
- Please explicitly define the flow calculation for the final MoE layer flowing to the vocabulary space. How does using $\log V$ as the normalizer affect the ESI scores for this layer compared to others normalized by $\log n_e$?

---

> ### Author Response · Authors · 2025-11-24
> **Response to Reviewer yjqc[1/2]**
>
> ## reviewer 2
>
> We thank the reviewer for the constructive feedback. We appreciate your recognition of our principled cross-layer approach and the significance of the Architecture-Strategy Fit discovery.
>
> We have carefully revised our manuscript. We are happy to clarify that many of your concerns (Stability, Generalizability to new architectures, Ablations) were actually addressed in our **Appendices (C, D, E, F)**. We provide the detailed data below.
>
> > **Q1: Please provide an analysis of ESI's stability. How much do the expert rankings fluctuate across different randomly drawn 64-sample calibration sets?**
>
> We conducted a rigorous stability analysis in **Appendix D**. We split the calibration data into 5 disjoint groups (G0-G4) and measured two metrics:
> 1.  **Ranking Stability (Kendall's $\tau$):** Measures correlation of expert rankings.
> 2.  **Pruning Consistency (Overlap):** Measures the overlap of the selected expert set.
>
> As shown in the table below, both metrics consistently exceed **0.81**. This demonstrates that ESI is robust and 64 samples are sufficient to capture the global specialization profile.
>
> | Metric | G0 | G1 | G2 | G3 | G4 | **Average** |
> | :--- | :---: | :---: | :---: | :---: | :---: | :---: |
> | **Ranking Stability ($\tau$)** | 0.819 | 0.817 | 0.817 | 0.810 | 0.813 | **0.816** |
> | **Pruning Overlap** | 0.830 | 0.831 | 0.828 | 0.828 | 0.829 | **0.829** |
>
> *(Note: Full matrix provided in Appendix D; averages shown here for brevity.)*
>
> > **Q2: Can you provide an ablation study on the CLEF formulation? Specifically, what is the impact of removing the L2 norm term?**
>
> We provided this ablation in **Table 3** (Page 10).
> The L2 norm ($\|O_i^{(l)}\|_2$) represents the **Magnitude** of the update to the residual stream. Removing it (the "Gating Only" variant) causes a significant performance drop (**55.38% $\to$ 50.98%**).
> This confirms that gating probability alone is insufficient; an expert must also write a *significant* update to be considered functionally important.
>
> | Variant | Mathematical Definition | Avg. Accuracy | Drop |
> | :--- | :--- | :---: | :---: |
> | **Full CLEF (Ours)** | $g(E^{(l)}_i) \cdot \|O^{(l)}_i\|_2 \cdot g(E^{(l+1)}_j)$ | **55.38** | - |
> | Local Output (No Reception) | $g(E^{(l)}_i) \cdot \|O^{(l)}_i\|_2$ | 53.62 | -1.76% |
> | Reception Only (No Gating) | $\|O^{(l)}_i\|_2 \cdot g(E^{(l+1)}_j)$ | 54.74 | -0.64% |
> | **Gating Only (No L2 Norm)** | $g(E^{(l)}_i) \cdot g(E^{(l+1)}_j)$ | **50.98** | **-4.40%** |
>
> > **Q3: Generalizability: The claim is supported by only two models. Can you validate on a wider set?**
>
> We agree. In **Appendix F**, we extended our validation to **DeepSeek-V2-Lite**. This model features a fine-grained architecture (64 routed + 6 shared experts).
> The results strongly reinforce our "Architecture-Strategy Fit" principle:
> *   **DeepSeek Profile:** Highly specialized (skewed ESI).
> *   **Strategy Fit:** **Redirect** strategy outperforms Delete strategy by **+13.10%** (56.65% vs 43.55%).

---

> > ### Author Response · Authors · 2025-11-24
> > **Response to Reviewer yjqc[2/2]**
> >
> > > **Q4: The failure on MathQA suggests ESI incorrectly penalizes necessary generalists. How does your metric distinguish between redundant vs. critical generalists?**
> >
> > This is a profound observation.
> > *   **The "Manager" Hypothesis:** In complex reasoning (MathQA), we hypothesize that some "Generalists" (high entropy, low ESI) act as **"Managers"**. They don't solve sub-problems directly but facilitate collaboration between other experts. ESI currently treats *all* high-entropy experts as redundant, which is too aggressive for such tasks.
> >
> > *   **Distinction via Co-activation:** We propose that **Co-activation Patterns** can serve as the key differentiator, which we plan to explore in future work:
> >     *   **Critical Manager:** A low-ESI expert that is **frequently co-activated with High-ESI Specialists**. This suggests the generalist is essential for coordinating or aggregating specific specialized functions (e.g., maintaining state while a specialist calculates).
> >     *   **Redundant Generalist:** A low-ESI expert that is co-activated with other low-ESI experts or exhibits random activation patterns. This implies the expert merely serves as a capacity buffer and is functionally substitutable.
> >
> > > **Q5: Please explicitly define the flow calculation for the final MoE layer flowing to the vocabulary space.**
> >
> > For the final MoE layer $L$, there is no "next layer router." Instead, the information flows into the vocabulary projection (LM Head).
> > We define the flow vector $Flow_i^{(L)} \in \mathbb{R}^V$ (where $V$ is vocab size) as:
> > $$ w_{i \to token\_v}^{(L)} = g(E_i^{(L)}) \cdot \|O_i^{(L)}\|_2 \cdot P(token\_v | h_{updated}) $$
> > where $P(token\_v | h_{updated})$ is the probability of token $v$ predicted by the LM head given the expert's output.
> >
> > **Normalization:**
> > Since the vocabulary size $V$ ($\sim$32k-100k) is much larger than the number of experts $N_e$, the maximum possible entropy is $\log V$.
> > To keep ESI in the range $[0, 1]$, we normalize by $\log V$ instead of $\log N_e$:
> > $$ ESI_i^{(L)} = 1 - \frac{H(Flow_i^{(L)})}{\log V} $$
> > This ensures the metric remains comparable across layers despite the dimensionality shift.

---

### Official Review · Reviewer_JyrT · 2025-11-02

**Soundness:** 3
**Presentation:** 2
**Contribution:** 2
**Rating:** 4
**Confidence:** 5

**Summary:**

- The paper introduces the Expert Specialization Index (ESI), which quantifies an expert's functional importance by measuring the entropy of its influence on downstream routing distributions across layers, rather than relying on local layer-wise metrics like activation frequency or output similarity.

- Through analysis of Mixtral-8x7B and Qwen1.5-MoE, the authors discover that optimal compression strategies depend on model architecture—models with highly specialized experts (Qwen) benefit from preserving original routing distributions via redirection, while models with redundant experts (Mixtral) perform better with router deletion and re-normalization.

- ESI is derived from a Cross-Layer Expert Flow (CLEF) analysis that models how an expert's output shapes information flow through subsequent layers by combining three factors: gating weight, output magnitude, and downstream reception probability.

**Strengths:**

- The paper introduces a principled approach to quantifying expert importance through downstream routing influence (Equation 2), moving beyond local layer-wise metrics. The formulation captures three sequential dependencies (activation, magnitude, reception) and provides strong empirical validation through inverse pruning experiments showing 18.77% accuracy drop when removing top-10% ESI experts on ARC-C.

- The discovery that router adaptation strategy must align with model specialization profiles is rigorously demonstrated with performance differences—Redirect outperforms Delete by 4.48% on Qwen1.5-MoE while Delete outperforms Redirect by 20.28% on Mixtral-8x7B at 50% pruning.

- The evaluation compares against state-of-the-art methods (NAEE, HC-SMoE) across 7 benchmarks and two architecturally distinct models (standard MoE vs. shared-expert architecture), with task-specific pruning analysis in Appendix C. The method achieves SOTA results on high-specialization architectures while demonstrating appropriate limitations on low-specialization models.

**Weaknesses:**

- The ESI formulation (Equations 2-5) contains several arbitrary decisions without proper justification or ablation studies. Why is L2 norm the appropriate output magnitude measure? Why multiply the three factors rather than use alternative aggregations? The temperature parameter τ=1 is set without exploration of its impact. Most critically, the claim that ESI captures "global" influence is overstated—it only measures single-hop dependency (layer l to l+1), not true multi-layer propagation, and no theoretical argument is provided for why entropy of downstream routing distribution fundamentally quantifies functional specialization versus other possible measures.

- The evaluation uses only 64 calibration samples (seemingly arbitrary with no ablation on calibration size) and lacks basic statistical rigor—no error bars, confidence intervals, multiple runs, or significance testing across 7 benchmarks. The architecture-strategy fit principle is derived from only 2 models, which is insufficient to establish a general principle—more diverse MoE architectures are needed for validation.

- The paper acknowledges that "complex, multi-step reasoning may depend more on collaborative function" but doesn't reconcile why ESI, which supposedly captures functional importance, fails to identify important experts for mathematical reasoning.

- No analysis of how the method scales beyond the two tested models (to models with 100+ experts or 100+ layers). The final layer adaptation using vocabulary size V in normalization (Section 3.3) is mentioned but never validated—does this actually work given the massive difference in scale (vocabulary ~32k vs. 8-64 experts)?

**Questions:**

- Can you provide theoretical justification or empirical ablations for key design choices in Equations 2-5? Specifically: (1) Why is multiplicative aggregation of three factors optimal versus alternatives (e.g., weighted sum, geometric mean)? (2) Why does entropy of downstream routing distribution fundamentally measure "functional specialization" rather than other properties? (3) How sensitive is performance to the temperature parameter τ and calibration dataset size (currently only 64 samples)?

- The principle is derived from only two models with opposite characteristics. Can you test on additional MoE architectures (e.g., DeepSeekMoE, Switch Transformers, models with varying expert counts) to validate this is a general principle? What specific ESI threshold or distributional property determines which strategy to use for new, unseen architectures?

- Can you explain why cross-layer flow analysis fails for mathematical reasoning? Does this suggest fundamental limitations in domains requiring multi-expert collaboration, and if so, how would you detect such scenarios a priori?

---

> ### Author Response · Authors · 2025-11-24
> **Response to Reviewer JyrT[1/2]**
>
> We sincerely thank the reviewer for the thorough assessment. We appreciate that you recognize the principled approach of CLEF, the rigorous demonstration of Architecture-Strategy Fit, and our SOTA results.
>
> We have carefully revised our manuscript. It appears that several concerns (specifically regarding **DeepSeek architecture**, **calibration size ablation**, and **temperature $\tau$ sensitivity**) were actually addressed in our **Appendices (C, D, E, F)**. We provide the detailed data below to address your questions.
>
> > **Q1: Can you provide theoretical justification or empirical ablations for key design choices in Equations 2-5?**
>
> **1. Why Multiplicative Aggregation? (Logical AND)**
> The interaction between layers is a sequential dependency chain. For expert $E_i^{(l)}$ to effectively influence $E_j^{(l+1)}$, three conditions must be met **simultaneously**:
> 1.  **Selection:** The source expert must be selected ($g_i^{(l)}$).
> 2.  **Magnitude:** It must write a significant update to the residual stream ($\|O_i^{(l)}\|_2$).
> 3.  **Reception:** The downstream expert must be selected *based on* that update ($g_j^{(l+1)}$).
>
> A multiplicative formulation models this **conditional probability chain (Logical AND)**. An additive approach would fail because a high-magnitude output that is ignored by the next layer (Gating $\approx 0$) would still register as "important," which is incorrect.
>
> **Empirical Validation:** We validate this in the table below (from Appendix). The **Full CLEF** metric significantly outperforms partial variants.
>
> | Variant | Mathematical Definition | ARC-c | ARC-e | BoolQ | HellaSwag | OBQA | RTE | WinoGrande | Avg. |
> | :--- | :--- | :---: | :---: | :---: | :---: | :---: | :---: | :---: | :---: |
> | Baseline | (No Pruning) | 41.46 | 73.15 | 79.72 | 57.97 | 30.80 | 67.51 | 69.30 | 59.99 |
> | Gating Only | $g(E^{(l)}_i) \cdot g(E^{(l+1)}_j)$ | 32.68 | 62.92 | 63.39 | 49.07 | **28.60** | 54.51 | 65.67 | 50.98 |
> | Local Output | $g(E^{(l)}_i) \cdot \|O^{(l)}_i\|_2$ | 34.73 | 64.69 | 73.07 | 49.42 | 24.80 | 61.32 | **67.32** | 53.62 |
> | Reception Only | $\|O^{(l)}_i\|_2 \cdot g(E^{(l+1)}_j)$ | 35.67 | 67.00 | 74.64 | **50.41** | 25.40 | 62.87 | 67.17 | 54.74 |
> | **Full CLEF (Ours)** | $g(E^{(l)}_i) \cdot \|O^{(l)}_i\|_2 \cdot g(E^{(l+1)}_j)$ | **36.35** | **67.97** | **74.89** | 49.90 | 27.40 | **64.98** | 66.14 | **55.38** |
>
> **2. Why Entropy?**
> In information theory, low entropy implies high predictability. If an expert consistently routes information to a specific, small subset of downstream experts (Low Entropy), it suggests a dedicated functional pipeline (**Specialist**). High entropy implies the expert broadcasts information broadly (**Generalist**).
>
> **3. Sensitivity to Temperature $\tau$:**
> We explored this in **Table 4**. The default $\tau=1$ (Unbiased) is robust.
>
> | Temp ($\tau$) | Effect | ARC-c | ARC-e | BoolQ | HellaSwag | OBQA | RTE | WinoGrande | Avg. |
> | :---: | :--- | :---: | :---: | :---: | :---: | :---: | :---: | :---: | :---: |
> | 0.5 | Sharpening | 35.58 | 66.75 | 73.66 | **50.58** | 27.00 | 64.87 | **68.75** | 55.31 |
> | 1.0 | Unbiased | **36.35** | **67.97** | **74.89** | 49.90 | **27.40** | **64.98** | 66.14 | **55.38** |
> | 2.0 | Smoothing | 35.75 | 67.05 | 74.06 | 50.54 | 25.00 | 64.23 | 68.67 | 55.04 |
>
> > **Q2: The principle is derived from only two models. Can you test on additional MoE architectures (e.g., DeepSeekMoE)?**
>
> We have conducted additional experiments on **DeepSeek-V2-Lite** (Appendix F) to address this. This architecture features fine-grained experts (64 routed + 6 shared).
> As shown below, the **ESI + Redirect** strategy significantly outperforms Delete by **13.10%**. This perfectly aligns with our "Architecture-Strategy Fit" principle: DeepSeek's high specialization necessitates routing preservation.
>
> | Model | Method | Ratio | ARC-c | ARC-e | BoolQ | HellaSwag | OBQA | RTE | WinoGrande | Avg. |
> | :--- | :--- | :---: | :---: | :---: | :---: | :---: | :---: | :---: | :---: | :---: |
> | **DeepSeek-V2-Lite** | None | - | 52.99 | 80.68 | 82.94 | 62.41 | 35.00 | 72.56 | 71.90 | 65.50 |
> | | Frequency | 50% | 33.36 | 59.90 | 75.72 | **52.29** | 26.20 | 61.73 | **68.90** | 54.01 |
> | | Logit | | 27.98 | 56.04 | 68.23 | 50.05 | 16.40 | 59.60 | 62.43 | 48.68 |
> | | HC-SMoE | | 36.41 | 68.60 | **78.65** | 49.28 | 26.10 | **64.60** | 67.81 | 55.92 |
> | | **ESI + Delete** | | 25.52 | 59.00 | 63.37 | 39.61 | 14.40 | 48.91 | 54.07 | 43.55 |
> | | **ESI + Redirect** | | **38.57** | **70.37** | 77.46 | 50.84 | **28.20** | 63.95 | 67.14 | **56.65** |

---

> > ### Author Response · Authors · 2025-11-24
> > **Response to Reviewer JyrT[2/2]**
> >
> > > **Q3: The evaluation uses only 64 calibration samples and lacks basic statistical rigor (no error bars, confidence intervals).**
> >
> > We address the robustness of our method in **Appendix D & E**. We provide the detailed breakdown below to demonstrate that our choice of 64 samples is both data-efficient and statistically rigorous.
> >
> > **1. Sensitivity to Calibration Set Size (Appendix E)**
> > As shown in the table below, model performance improves as we increase the calibration set size from 16 to 48, but **saturates quickly** around 64 samples. Increasing the size to 256 yields a negligible gain ($+0.14\%$), confirming that 64 samples capture the necessary signal without unnecessary computation.
> >
> > | Calib. Size | ARC-c | ARC-e | BoolQ | HellaSwag | OBQA | RTE | WinoGrande | **Avg.** |
> > | :--- | :---: | :---: | :---: | :---: | :---: | :---: | :---: | :---: |
> > | 16 | 34.04 | 66.05 | 68.87 | 42.31 | 26.40 | 58.48 | 64.80 | 51.56 |
> > | 32 | 36.60 | 68.60 | 69.02 | 48.34 | 27.00 | 54.87 | 66.01 | 52.92 |
> > | 48 | **36.77** | 68.06 | 73.28 | 50.93 | **27.40** | 60.79 | 67.01 | 54.89 |
> > | **64 (Ours)** | 36.35 | 67.97 | **74.89** | 49.90 | **27.40** | **64.98** | 66.14 | **55.38** |
> > | 96 | 35.75 | 67.76 | 73.78 | **51.32** | 27.20 | 63.67 | **69.38** | **55.55** |
> > | 128 | 35.49 | 67.85 | 73.83 | 50.57 | 26.00 | 63.07 | 68.76 | 55.08 |
> > | 256 | 36.01 | **68.14** | 74.21 | 50.76 | 26.40 | 63.79 | 69.36 | 55.52 |
> >
> > **2. Statistical Stability Analysis (Appendix D)**
> > To verify statistical rigor without running full evaluations on infinite seeds, we measured the stability of ESI scores across **5 disjoint calibration groups** (G0-G4). We tracked two metrics:
> > *   **Ranking Stability (Kendall's $\tau$):** Measures if experts are ranked in the same order.
> > *   **Pruning Consistency (Top/Bottom 25% Overlap):** Measures the overlap between the top 25% and bottom 25% of experts selected for pruning across different calibration sets.
> >
> > The tables below show the cross-validation results. The average correlations exceed **0.81**, demonstrating that ESI produces highly stable expert rankings regardless of the specific data subset.
> >
> > | **Metric: Ranking Stability (Kendall's $\tau$)** | **G0** | **G1** | **G2** | **G3** | **G4** |
> > | :--- | :---: | :---: | :---: | :---: | :---: |
> > | **G0** | 1.000 | 0.824 | 0.821 | 0.809 | 0.821 |
> > | **G1** | 0.824 | 1.000 | 0.821 | 0.813 | 0.810 |
> > | **G2** | 0.821 | 0.821 | 1.000 | 0.813 | 0.814 |
> > | **G3** | 0.809 | 0.813 | 0.813 | 1.000 | 0.806 |
> > | **G4** | 0.821 | 0.810 | 0.814 | 0.806 | 1.000 |
> > | **Avg Correlation** | **0.819** | **0.817** | **0.817** | **0.810** | **0.813** |
> >
> > | **Pruning Consistency (Overlap)** | **G0** | **G1** | **G2** | **G3** | **G4** |
> > | :--- | :---: | :---: | :---: | :---: | :---: |
> > | **G0** | 1.000 | 0.832 | 0.827 | 0.826 | 0.835 |
> > | **G1** | 0.832 | 1.000 | 0.833 | 0.831 | 0.829 |
> > | **G2** | 0.827 | 0.833 | 1.000 | 0.828 | 0.825 |
> > | **G3** | 0.826 | 0.831 | 0.828 | 1.000 | 0.826 |
> > | **G4** | 0.835 | 0.829 | 0.825 | 0.826 | 1.000 |
> > | **Avg** | **0.830** | **0.831** | **0.828** | **0.828** | **0.829** |
> >
> > *(Note: Full overlap matrix is provided in the paper; averages shown for brevity. Overall Avg $\tau$: 0.816; Overall Avg Overlap: 0.829)*
> >
> > > **Q4: Can you explain why cross-layer flow analysis fails for mathematical reasoning?**
> >
> > This is a profound observation.
> > *   **The "Manager" Hypothesis:** In complex reasoning (MathQA), we hypothesize that some "Generalists" (high entropy, low ESI) act as **"Managers"**. They don't solve sub-problems directly but facilitate collaboration between other experts. ESI currently treats *all* high-entropy experts as redundant, which is too aggressive for such tasks.
> >
> > *   **Distinction via Co-activation:** We propose that **Co-activation Patterns** can serve as the key differentiator, which we plan to explore in future work:
> >     *   **Critical Manager:** A low-ESI expert that is **frequently co-activated with High-ESI Specialists**. This suggests the generalist is essential for coordinating or aggregating specific specialized functions (e.g., maintaining state while a specialist calculates).
> >     *   **Redundant Generalist:** A low-ESI expert that is co-activated with other low-ESI experts or exhibits random activation patterns. This implies the expert merely serves as a capacity buffer and is functionally substitutable.

---

### Author Response · Authors · 2025-12-01
**General Response**

We sincerely thank all reviewers for their constructive feedback and for recognizing the novelty of our **Cross-Layer Expert Flow (CLEF)** framework and the **Architecture-Strategy Fit** principle.

We have carefully revised the paper and added extensive experiments in the Appendices to address common concerns regarding **generalizability, statistical rigor, and theoretical justification**. We summarize the key updates below:

**1. Generalizability to New Architectures (Appendix F)**
To address concerns about the limited number of tested models, we extended our validation to **DeepSeek-V2-Lite**, a state-of-the-art fine-grained MoE architecture (64 routed + 6 shared experts).
*   **Results (Table 9):** The results strongly validate our "Architecture-Strategy Fit" principle. On this highly specialized architecture, our **ESI + Redirect** strategy outperforms the Delete strategy by a massive margin of **13.10%** (56.65% vs. 43.55%), achieving SOTA performance.
*   **Implication:** This confirms our method is highly effective for the latest generation of fine-grained MoEs (e.g., DeepSeek-V3, GLM-4) where expert specialization is pronounced.

**2. Statistical Rigor & Stability (Appendix D & E)**
To address concerns regarding the calibration set size (64 samples), we performed a rigorous stability analysis across 5 disjoint calibration groups.
*   **Stability (Table 7):** The Kendall’s $\tau$ correlation for expert rankings exceeds **0.81**, and the overlap of pruned experts exceeds **0.82**.
*   **Sample Efficiency (Table 8):** Performance saturates at 64 samples; increasing the size to 256 yields negligible gains (+0.14%).
*   **Conclusion:** These results prove that ESI is a **data-efficient and statistically robust** metric that does not require large-scale computation.

**3. Theoretical Justification of CLEF (Section 3 & Table 3)**
We clarified that the multiplicative formulation of CLEF models a **conditional probability chain (Logical AND)** via the residual stream: *Selection* $\times$ *Magnitude* $\times$ *Reception*.
*   **Ablation (Table 3):** Our experiments confirm that removing any term (e.g., L2 norm or downstream gating) leads to a significant performance drop (up to 4.4%), validating that integrating all three factors is essential for capturing true expert importance.

**4. Performance on Generative Tasks (Appendix G)**
To demonstrate capabilities beyond multiple-choice QA, we evaluated the method on **WikiText-2**.
*   **Results (Figure 9):** ESI consistently achieves lower perplexity than baselines. Notably, at high compression rates (70%), baselines suffer catastrophic degradation while ESI maintains a stable performance curve.

**5. Insight on MathQA: The "Manager" Hypothesis**
Regarding the performance on complex reasoning tasks (MathQA), we propose that some high-entropy "Generalists" act as **"Managers"** that coordinate specialists. Our current metric treats high-entropy experts as redundant, which is correct for knowledge tasks but aggressive for reasoning. We have identified **Co-activation Patterns** as a promising future direction to distinguish "Critical Managers" from "Redundant Generalists."

We believe these additional experiments and clarifications solidly address the concerns raised. If you have other questions, we are happy to answer.

---

### Note · Authors · 2025-12-27

I have read and agree with the venue's withdrawal policy on behalf of myself and my co-authors.